# FORMAL THEOREM PROVING BY REWARDING LLMS TO DECOMPOSE PROOFS HIERARCHICALLY

## ABSTRACT

Mathematical theorem proving is an important testbed for large language models' deep and abstract reasoning capability. This paper focuses on improving LLMs' ability to write proofs in formal languages that permit automated proof verification/ evaluation. Most previous results provide human-written lemmas to the theorem prover, which is an arguably oversimplified setting that does not sufficiently test the provers' planning and decomposition capabilities. Instead, we work in a more natural setup where the lemmas that are directly relevant to the theorem are not given to the theorem prover at test time. We design an RL-based training algorithm that encourages the model to decompose a theorem into lemmas, prove the lemmas, and then prove the theorem by using the lemmas. Our reward mechanism is inspired by how mathematicians train themselves: even if a theorem is too challenging to be proved by the current model, a reward is still given to the model for any correct and novel lemmas that are proposed and proved in this process. During training, our model proves 37.7% lemmas that are not in the training dataset. When tested on a set of holdout theorems, our model improves the pass rate from 40.8% to 45.5% compared with the supervised fine-tuned (SFT) model.

## 1 INTRODUCTION

The reasoning abilities of large language models (LLMs) are a significant marker of artificial intelligence and critical for complex and safety-sensitive applications, e.g., medical diagnosis (Fleming et al., 2023; Singhal et al., 2023), legal document review (Guha et al., 2024),online tutoring (Ruan et al., 2024; Wang & Demszky, 2023). Yet recent studies highlight the limited performance of LLMs on reasoning tasks (e.g., Mündler et al. (2023); Valmeekam et al. (2023) and references therein).

Automated theorem proving by LLMs is an excellent reasoning task that abstracts away the need for numerical manipulation or tool use (e.g., using a calculator) and allows for precise evaluation of correctness with an automatic verifier (such as Isabelle (Nipkow et al., 2002) and Lean (De Moura et al., 2015)) even without ground truth. Thanks to tools such as Sledgehammer (Paulsson & Blanchette, 2012) that can automatically complete low-level details, the granularity of formal proofs is similar to natural language proofs (see Fig. 1 (Left) for an illustrative example). Note that verifying a proof is fundamentally much easier than generating the proof.[1] Thus, learning to prove theorems from verifiers' supervision is reminiscent of weak-to-strong generalization (Burns et al., 2023).

Previous results in this area largely focus on the setting where the theorem prover can use all the lemmas in the formal proof library, including those particularly written to decompose a specific theorem's proof (Jiang et al., 2021; Polu & Sutskever, 2020). This setting arguably oversimplifies the problem and doesn't sufficiently test the models' planning and decomposition capabilities, and it is unclear whether the resulting models can be used to prove new theorems from scratch when such lemmas are not available at test time. Instead, we work in a more natural setup where the theorem prover needs to propose and prove lemmas to decompose the proof hierarchically itself (see Section 2 for more details). In Section 4.2, we demonstrate that this task is much more challenging.

In addition, most existing proof-generation algorithms leverage the formal verifier by (a) providing the verifier's current proof state to the LLMs step-by-step, and (b) using best-first search algorithms such as A$^\star$ to build a multi-step proof from many LLM-generated steps (Han et al., 2021; Jiang et al.,

---

[1]The former is in P whereas the latter is undecidable in the worst case Church (1936); Turing et al. (1936).

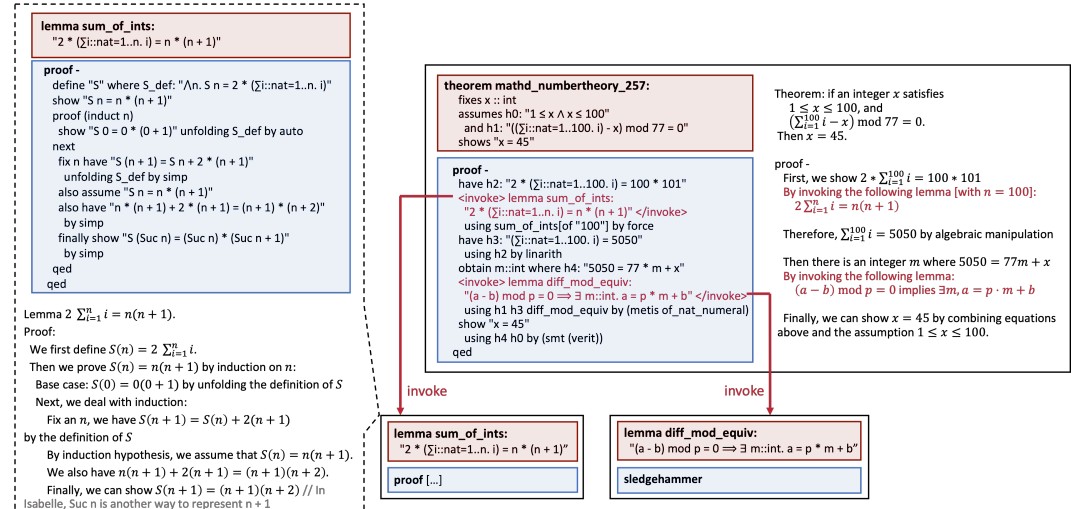

Figure 1: **Left**: in the dashed callout block, we show an example of an Isabelle proof and its explanation in natural language. **Right**: an example of a proof tree. The two child nodes correspond to the two new lemmas proposed in the proof of the root node.

2022b). The major challenge of these methods is the high computation cost incurred in both (a) and (b) because (a) requires re-running LLMs on a different context that consists of a verifier's (long) proof state at every step, and (b) requires generating many proof steps first and then select the best ones. Consequently, the best method along this line of research requires more than 1k GPU days with A100s to train a model with 600M parameters (Lample et al., 2022), whereas our method only takes less than 36 GPU days to train a 7B parameter model.

To address these issues, we design a method, Proof Decomposer (ProD), which uses LLMs to hierarchically propose and prove new lemmas and generate complete proofs directly without searching. We augment the formal proofs syntax so that the model can propose new lemmas by including their statements during the proof, and prove these lemmas separately. Hence, a complete proof of a theorem forms a tree structure where the child nodes are the lemmas proposed in the proof of the parent node (Fig. 1 (Right)), and the theorem is considered proven only if all the proofs in the tree are correct.

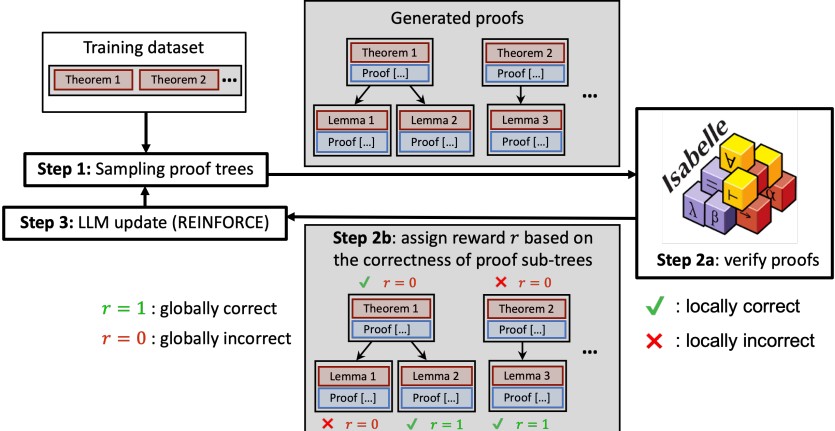

Figure 2: Illustration of our algorithm Proof Decomposer, *ProD-RL*. In step 2b, the statement is *locally correct* if it is proved correctly using the proposed lemmas, and it is *globally correct* if all the proposed lemmas are also proved correctly. As an important feature of our algorithm, even if a theorem (Theorem 1) is not proved by the model because some lemmas (Lemmas 1) are not proved, we still train on the correct lemma (Lemma 2) by setting its reward $r = 1$.

We train our models with reinforcement learning (RL) in a way that somewhat imitates a mathematician's process: we reward correct partial proofs (i.e., proof sub-trees) even if the original theorem (i.e., the root node) is not proved entirely. Since our model can generate and prove novel lemmas during training, it can still make progress even if the theorem is too challenging. This is reminiscent of how mathematicians prove standalone lemmas which make progress towards solving an open problem. We illustrate our algorithm in Fig. 2, and defer the details to Section 3.2.

We test our model ProD-RL by generating proof trees on holdout theorems that the model is never trained on, and we show that our model ProD-RL outperforms several other baselines. Compared with the supervised fine-tuned (SFT) model on the same training set , our model improves the pass rate from 40.8% to 45.5% on the holdout test set, whereas vanilla reinforcement learning without lemma proposals during training does not improve the corresponding SFT model (see Section 4.3). This is partly because our method encourages the model to propose and prove additional lemmas — in fact, 37.7% of the lemmas proved during training are not in the dataset. As a result, the model still improves even if it is already fine-tuned on the same dataset with human-written ground-truth proofs.

## 2 SETUP

**Conditional proofs.** We use the term *conditional proof* to denote a proof that, in addition to the standard formal proof syntax, can propose new lemmas by enclosing their statements by <invoke> and </invoke> tokens (examples shown in the blue boxes of Fig. 1). In particular, a conditional proof has the following format:

$$t_1 \text{ <invoke> } l_1 \text{ </invoke> } t_2 \text{ <invoke> } l_2 \text{ </invoke> } \cdots t_k \text{ <invoke> } l_k \text{ </invoke> } t_{k+1}$$

where $t_1, \cdots, t_{k+1}$ denote proof segments in the original formal proof syntax (see e.g., Fig. 1, proof texts in black), and $l_1, \cdots, l_t$ denote proposed lemma statements (see e.g. Fig. 1, proof texts in red).[2]

**Proof tree nodes.** With the proposed lemmas, a complete proof forms a tree structure (as shown in Fig. 1). A node in a proof tree is a tuple of premises, context, a theorem statement, and a conditional proof. Premises represent the lemmas that are treated as common knowledge, which are typically not directly relevant to the proof. We allow the model to directly use them in the proof so that it does not have to repetitively prove all the fundamental facts, such as properties of continuous functions and natural numbers. Context represents the necessary contents to prepare the theorem statement, such as the definition of specific objects/functions. We use the context as part of the prompt for the LLMs to generate proofs, and to configure the proof verifier to check the generated proofs.

**Correctness of conditional proofs and proof trees.** A proof tree node $n$ with conditional proof $t_1$ <invoke> $l_1$ </invoke> $t_2$ <invoke> $l_2$ </invoke> $\cdots t_k$ <invoke> $l_k$ </invoke> $t_{k+1}$ is *locally correct* if, after adding $l_1, \ldots, l_k$ to the set of premises, $t_1 \ldots t_{k+1}$, is a proof to the statement of $n$ that is accepted by the formal verifier under the context of $n$.

We consider a proof tree valid if, for every node, each of its child nodes corresponds to one proposed lemma and shares the same premises and context with its parent node. A tree node $n$ is *globally correct* with respect to a given set of tree nodes $N$ if we can construct a valid proof tree with root $n$ using the locally correct tree nodes in $N$. We use this more flexible definition of global correctness since if we generate more than one proof tree per theorem, we may mix their locally correct nodes to form a globally correct proof.

Global correctness corresponds to the standard notion of correctness (i.e., whether the theorem is proved), and local correctness is a weaker concept, referring to the correctness of conditional proofs assuming the proposed lemmas. When a tree node is globally correct, we can construct a complete proof to its statement that is acceptable by the formal proof verifier — first, we build a valid proof tree from the locally correct subset of $N$, and then list all the statements and their corresponding conditional proofs in a child-first order and remove all the lemma proposal steps (since the proposed lemmas and their proofs will be already listed in the proof text according to child-first order).

---

[2]In this paper, we use the terms 'lemma' and 'theorem' relatively — theorem refers to the statement that we are currently focusing on, and lemma refers to the statement proposed during the proof. In other words, there is no fundamental difference between a lemma and a theorem.

**Dataset construction.** We construct the datasets by parsing raw proof-library files into tuples of the form (premises, context, statement, conditional proof).

In particular, we first segment each of the files into blocks $c_1\ s_1\ p_1\ \cdots\ c_l\ s_l\ p_l$ where the $s_i$ are theorem statements, the $p_i$ are the corresponding proofs, and the $c_i$ are the file contents between proofs, such as object definitions and local assumptions. Next, we build proof trees from each segmented file by iteratively removing $(s_i, p_i)$ pairs from the file if the theorem $s_i$ is not referred to in the remaining file contents (in other words, in the first iteration we peel off the root nodes of the proof trees from the file, and then the nodes in the next level, etc.). Note that some theorems cannot be peeled off by this process because they are referred to in some file content $c_j$ (e.g., lemmas used to instantiate local objects). We use $\mathcal{T}_{\text{tree}}$ to denote the subset of theorems peeled off during the process.

For every theorem $s_i$, we construct an example where the context is the concatenation of $\{c_j : j < i\}$ and $\{s_j : j < i, s_j \notin \mathcal{T}_{\text{tree}}\}$ in the order they appear in the file. That is, we exclude all the lemmas that are ever peeled off — the remaining lemmas are included in the context.

To construct the conditional proof of theorem $s_i$, we add the proposed lemma statements to the original proof $p_i$. In particular, we split the proof $p_i$ into steps $t_1, \cdots, t_k$ using the formal language parser. Then for every step $t_j$ that uses lemmas $l_{j,1}, \ldots, l_{j,n_j}$ from $\mathcal{T}_{\text{tree}}$, we insert the statements of these lemmas enclosed by the <invoke> and </invoke> tokens, denoted by $\zeta_j = $ <invoke> $l_{j,1}$ </invoke> $\cdots$ <invoke> $l_{j,n_j}$ </invoke>, into the proof right before $t_j$. In other words, the conditional proof is the concatenation $\zeta_1\ t_1\ \cdots\ \zeta_k\ t_k$. Similar to Jiang et al. (2022b), we use Sledgehammer, a premise selection tool that automatically searches for proofs to the current goal, to replace proof steps that are originally generated by it (see Section A.3 for more details) so that the mode can focus less on the tedious low-level details.

The premises are all the theorems from *predecessor* files, which are typically not directly relevant to the theorem (otherwise they will be stated in the same file). Theorems in the premise set can be used directly in the proof, or they can be selected by Sledgehammer to search for proof steps. In our implementation, the premises are implicitly defined by the dependency graphs of the files.

We split the training and test set (AFP test) based on the dependency of the files in the proof library so that the examples in the training set never refer to any files in the test set (see Section 4.2 for details). We also construct an additional test set, AFP 2023, by parsing AFP files submitted after the knowledge cutoff date of the Llemma model (April 2023) to eliminate potential data leakage issues. Compared with prior works (First et al., 2023; Jiang et al., 2021), the two major differences in our setup are the availability of lemmas from the same file and the training/test split. In Section 4.2, we discuss and test their effects in detail.

Finally, to construct the SFT dataset, for each example in the training set, if its conditional proof proposes at least one lemma, we create an augmented example by moving the proposed lemmas from the conditional proof into the context — this augmented example does not propose new lemmas and is always locally correct.

## 3 METHODS

In this section, we first describe how to use LLMs to generate proof trees, and then introduce our reinforcement learning method (ProD-RL) that rewards the model to decompose proofs hierarchically.

### 3.1 GENERATING PROOF-TREES USING LLMS

To generate proof trees using an autoregressive model $\pi_\theta$, we need to first fine-tune the model to follow a specific format:

(a) the input $x$ to the model $\pi_\theta$ is the concatenation of a context and a theorem statement, and

(b) the expected output $y$ of the model is a special token $t_0$ followed by a conditional proof, where $t_0$ is either <use_invoke> or <no_invoke>, denoting whether the following conditional proof should propose new lemmas.

We introduce the special token $t_0$ before a conditional proof so that we can increase the probability of the <use_invoke> token during RL to let the model propose more lemmas for better exploration.

We summarize our proof-tree generation algorithm in Alg. 1. Given a theorem statement $s$ and the corresponding context $c$, we first sample from $\pi_\theta$ autoregressively starting with the prompt $x = c\ s$, and ideally the model outputs a special token $t_0$ followed by a conditional proof $\rho$ (Line 3). Next, we parse the conditional proof $\rho$ and collect the proposed lemmas $l_1, \cdots, l_k$ for the next round of generation (Line 5). We force the model to generate conditional proofs without proposing new lemmas at a certain depth so that the proof tree doesn't grow indefinitely, which can be implemented easily by replacing the prompt with $x' = c\ s$ <no_invoke> (Line 6).

---

**Algorithm 1** Generate proof trees (test time)

---

1: **Inputs:** Model $\pi_\theta$, a set of contexts and statements $G_0 = \{(c_i, s_i)\}_i$, maximum depth $d$.
2: **for** $\iota \leftarrow 0, 1, \cdots, d-1$ **do**
3:     Sample proofs $(t_{0,i}\ \rho_i) \sim \pi_\theta(\cdot \mid c_i\ s_i)$ for lemmas $(c_i, s_i)$ in $G_\iota$, where $t_{0,i}$ is the token representing whether the proof should use invoke and $\rho_i$ is the conditional proof.
4:     $P_\iota \leftarrow \{(c_i, s_i, \rho_i) \mid \forall(c_i, s_i) \in G_\iota\}$.
5:     Collect proposed lemmas (a conditional proof $\rho_i$ might propose more than one lemma $l_j$):
$$G_{\iota+1} \leftarrow \{(c_i, l_j) \mid (c_i, s_i, \rho_i) \in P_\iota \text{ and } l_j \text{ is proposed in } \rho_i\}.$$
6: Sample proofs $\rho_i \sim \pi_\theta(\cdot \mid c_i\ s_i$ <no_invoke>$)$ for $(c_i, s_i)$ in $G_d$.     ($\triangleright$) Truncate at depth $d$.
7: $P_d \leftarrow \{(c_i, s_i, \rho_i) \mid \forall(c_i, s_i) \in G_d\}$.
8: **Return** $\cup_{\iota=0}^d P_\iota$.

---

### 3.2 REINFORCEMENT LEARNING WITH LEMMA PROPOSAL

Our reinforcement learning method is illustrated in Fig. 2. We start with a supervised fine-tuned model so that it can generate conditional proofs in the desired format. Then at every round, we randomly sample a batch of examples $D$ from the training dataset and perform the following steps.

**Step 1: Generate proofs.** We first generate proof trees for every theorem in $D$. For better exploration, we use a modified version of Alg. 1 (shown in Alg. 2 of Appendix A.1) with the following differences:

(a) for the theorems where the probability $\pi_\theta($<use_invoke> $\mid x)$ is among the top 50% in the batch, we will force the model to generate conditional proofs with $t_0 = $ <use_invoke>. Otherwise, we sample $t_0$ according to the probability of $\pi_\theta(\cdot \mid x)$, and

(b) for every theorem where the model generates a conditional proof with new lemmas, we also let the model generate another conditional proof without proposing lemmas. If any of these two conditional proofs is globally correct, it can be used to construct proof trees for other theorems.

**Step 2: Determine the reward of an example.** In this step, we first check the local correctness of each conditional proof using the formal verifier (Step 2a in Fig. 2).

In addition to the verifiers' output, we apply two filters to help train the model: (a) we filter out trivial lemma proposals — if a proposed lemma directly implies the theorem (e.g., if the proposed lemma has exactly the same statement as the theorem), we simply discard this example, and (b) we remove unnecessary lemma proposals — if the conditional proof is still correct after removing all the references to a proposed lemma, we remove this lemma from the conditional proof.

We then determine the global correctness of the generated proofs. Finally, we assign a binary reward $r(c, s, \rho)$ to each tree node with context $c$, statement $s$, and conditional proof $\rho$ based on its global correctness (Step 2b in Fig. 2).

**Step 3: Update the model by REINFORCE.** In this step, we first construct a training dataset consisting of examples with format (prompt, target, weight) from the conditional proofs collected in Step 1, and then update the model $\pi_\theta$ using the weighted cross-entropy loss.

For each generated conditional proof, we add one example to the training dataset where the prompt is the context concatenated with the theorem statement, and the target is the conditional proof prepended by the <use_invoke> or <no_invoke> token. Note that the reward of a conditional proof depends not only on the correctness of the conditional proof, but also on the correctness of the proposed lemmas.

To reduce the variance of our gradient updates, we train a value function $V_\phi$ that predicts the expected reward of the current policy on a given proof tree node (i.e., $V_\phi \approx \mathbb{E}_{\rho \sim \pi(\cdot|c,s)}[r(c, s, \rho)]$). The weight of an example is the product of the value function's outputs on invoked lemmas multiplied with a length penalty to incentivize shorter proofs — for a proof tree node with conditional proof length $h$ and proposed lemmas $l_1, \cdots, l_k$, the weight $w$ of this example is $w = \gamma^h \prod_{i=1}^{k} V_\phi(l_i)$ with discount factor $\gamma \in (0, 1)$, or $w = 0$ if the proof tree node is not locally correct.

To simplify the implementation, we train the value function to predict two special tokens, <true> and <false>, conditioned on the context and theorem statement. Let $p_t$ be the probability of the <true> token conditioned on the context and theorem statement, and $p_f$ the probability of the <false> token. The output of the value function is then $p_t/(p_t + p_f)$.

As done for the SFT dataset, we add one augmented example by moving the proposed lemma from the conditional proof to the context for any locally correct conditional proof with new lemmas. We also add examples constructed from the human-written conditional proofs (i.e., the ground-truth proofs) of theorems in the batch $D$. In addition, we use a replay buffer to stabilize the training.

**Remarks.** Note that we update the model using partial proofs (i.e., proof sub-trees) even if the original theorem from the dataset (i.e, the root of the proof tree) is not proved. Hence, our method can also be viewed as an instantiation of hindsight experience replay (Andrychowicz et al., 2017), where the hindsight trajectories are correct proof sub-trees.

Our algorithm is also closely related to expert iteration. In our notation, expert iteration is equivalent to using a binary weight $w = \mathbb{I}$ [the proof tree node is *globally* correct] .

## 4 EXPERIMENTS

This section presents our experimental results. We first list additional experiment details (Section 4.1) and then compare our setup with prior works (Section 4.2). Finally, we show our main results in Section 4.3 and examples of proposed lemmas in Section 4.4.

### 4.1 EXPERIMENT DETAILS

**Proof verification software.** We use Isabelle (Nipkow et al., 2002) as our proof verification software since the proofs are declarative and human-readable without knowing the verifier's proof state, and we use PISA (Portal to ISAbelle, Jiang et al. (2021)) to interact with Isabelle. To check whether a proof tree node is locally correct, we import all the theorems from its premises, move each of the proposed lemmas from the conditional proof to the context, and then add a fake proof indicated by the keyword 'sorry' to every lemma statement in the context (In Isabelle, 'sorry' will register the statement as a fact even without any actual proof.) The remaining proof steps will follow the original Isabelle syntax, and we can check their correctness directly. We set a 10s timeout for each proof step.

**Datasets.** Our SFT dataset consists of theorems from Archive of Formal Proof[3] (AFP, retrieved on 2022-12-06) and Isabelle built-in files (such as HOL which contains the theorems that define natural numbers, etc.). The resulting dataset contains 312k examples.

For the test datasets AFP test and AFP 2023, we only keep the theorems in $\mathcal{T}_{tree}$. To construct the test set AFP 2023, we parse the AFP files submitted after the knowledge cutoff date of our pretrained model (April 2023) to eliminate possible data leakages.[4] The AFP test set contains 4.3k theorems and the AFP 2023 test set 2k theorems.

**Testing setup.** To measure the performance of the models, we sample $k$ proof trees per theorem independently on the test set and report the pass@k performance (that is, a theorem is proved if at least one of the conditional proofs is globally correct with respect to all the generated tree nodes). When generating proofs, we use temperature 0.7 and truncate the context to only include the last 1k tokens. The proof trees are truncated at depth 2.

---

[3]https://www.isa-afp.org/
[4]Here we use the archive of AFP retrieved on 2023-11-22.

**Supervised fine-tuning.** We start from the Llemma 7b model (Azerbayev et al., 2023) and fine-tune the model for 2 epochs with the standard cross entropy loss.[5] On theorems from AFP, we compute the loss only on the special token and the proof, but not on the context and statement. On Isabelle built-in theorems, we compute the loss on the statement to help the model internalize basic facts. We use the AdamW optimizer (Loshchilov & Hutter, 2018) with linear warmup, constant learning rate, peak learning rate 1e-5, macro batch size 128, and context window 2048.

**Reinforcement learning.** The dataset we use for the reinforcement learning stage is $\mathcal{T}_{\text{tree}}$, the set of theorems that are iteratively peeled off when parsing AFP files — $\mathcal{T}_{\text{tree}}$ contains 104k examples. We first train the model with 1 epoch of supervised fine-tuning, and then run RL for 20 rounds with a batch of 5k random examples per round. We truncate the proof tree at depth 3 and sample with temperature 0.7 during training for better exploration. We use the same hyperparameters as the SFT stage to update the policy $\pi_\theta$, We initialize the value function $V_\phi$ with a Llemma 7b model fine-tuned on our SFT dataset.

### 4.2 COMPARISON OF OUR NEW SETUP WITH PRIOR WORKS

In this section, we concretely compare our new setup with prior works (First et al., 2023; Jiang et al., 2021). Recall that there are two main differences in how we process our dataset:

(a) we split the train/test set based on file dependencies so that no theorems in the test set are referred to in the training set, whereas PISA splits theorems randomly, and

(b) when testing a proof, we remove certain lemmas from the context.

To show that our setup is indeed more challenging, we first construct datasets formatted similarly to those in (First et al., 2023). Specifically, we parse the AFP files into examples using the method described in Section 2, with the only exception being that all human-written lemmas are kept in the context. Then we select a subset of theorems as the test set $D_{\text{test}}^{\text{w/l}}$ based on the dependencies of the AFP files, so that the examples in the test set are never used by the remaining theorems (see Section A.3 for more details). Then we split the remaining examples randomly into training and validation datasets, denoted by $D_{\text{train}}^{\text{w/l}}$ and $D_{\text{val}}^{\text{w/l}}$ respectively. We use $D_{\text{test}}^{\text{w/o l}}$ to denote the test dataset of our setup where the lemmas are removed from the context. The validation dataset $D_{\text{val}}^{\text{w/l}}$ mimics prior works' setup (First et al., 2023; Jiang et al., 2021), and $D_{\text{test}}^{\text{w/l}}$ is an interpolation between prior works' setup and our setup. Table 1 shows the performance of the model with supervised fine-tuning on $D_{\text{train}}^{\text{w/l}}$ and all Isabelle built-in theorems. The results suggest that both features of our setup, removing the lemmas and splitting the training/test set by file dependencies, increase the difficulty of the task.

Table 1: Pass rate on different dataset formats and partitions of the SFT model trained on the $D_{\text{train}}^{\text{w/l}}$. The validation dataset $D_{\text{val}}^{\text{w/l}}$ mimics the test setup of prior works, and our setup is $D_{\text{test}}^{\text{w/o l}}$ where the same model performs much worse. The results suggest that our setup is indeed more challenging.

| Test setup | $D_{\text{val}}^{\text{w/l}}$: w/ lemmas split randomly | $D_{\text{test}}^{\text{w/l}}$: w/ lemmas split by dependency | $D_{\text{test}}^{\text{w/o l}}$: w/o lemmas split by dependency |
|---|---|---|---|
| pass@4 | 45.7 | 39.7 | 35.7 |

### 4.3 MAIN RESULTS

This section reports the models' pass@k performance on AFP test and AFP 2023 test datasets. Recall that the theorems in the AFP 2023 test set were submitted after we retrieved the training set, and theorems in AFP test are selected based on the dependencies of the AFP files. Hence, theorems in the test sets are not used in any proofs from the training set. In other words, we do not train the model on test datasets using reinforcement learning. Instead, we test whether ProD-RL is a fundamentally better model when tested on new theorems.

As a baseline method, we train a model on a variant of the SFT dataset where all lemmas are kept in the context. It can be seen as a reproduction of First et al. (2023) with a slightly different way to obtain

---

[5]In our preliminary experiments, we observe that the model overfits after 2 epoch

the context — First et al. (2023) includes all the file content before the statement of the theorem, whereas we only keep the statement of previous lemmas. We also run reinforcement learning on the same RL dataset as our method (see Section A.2 for more details).

Table 2: Pass@16 of different models on AFP test sets. Our model with reinforcement learning (ProD-RL) improves upon the SFT model and outperforms baseline methods.

| Test set | SFT w/o lemma proposal | RL w/o lemma proposal | ProD-SFT | ProD-RL |
|---|---|---|---|---|
| AFP test | 43.4 | 42.4 | 40.8 | **45.5** |
| AFP 2023 | **39.4** | 37.7 | 36.5 | **39.5** |

Table 2 shows the performance of our model on the AFP test sets. For a fair comparison, the baseline models are tested in our new setup without human-written lemmas. Note that the SFT model without lemma proposal outperforms the SFT model with lemma proposal. We hypothesize that it is because proposing correct lemmas itself is challenging, which distracts the model from learning to generate direct proofs. However, RL with lemma proposal improves the SFT model and outperforms others because the model proposes and proves additional lemmas that are not in the training dataset, whereas RL without lemma proposal yields no improvement.

In Fig. 3, we plot the pass rates with different numbers of samples per theorem on both AFP test and AFP 2023. Fig. 3 shows that on AFP test, the ProD-RL model significantly improves upon baseline methods as well as the ProD-SFT. However, on AFP 2023, the improvement is minor over SFT w/o lemma proposal, while ProD-RL still outperforms ProD-SFT. The results suggest that the baseline methods are more robust to heavier distribution shifts, while our method has a larger improvement when the test distribution is closer to the training distribution.

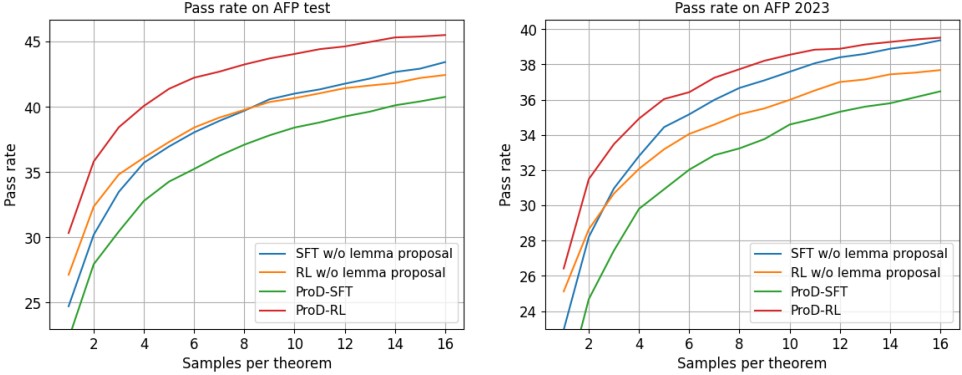

Figure 3: The pass rate of different models on AFP test (Left) and AFP 2023 (Right) test sets. Our RL model improves upon the SFT model whereas without proposing new lemmas (RL w/o lemma proposal), we do not observe any improvement.

In Fig. 4, we decompose the proved theorems by the depth of their ground-truth proofs (shown on the $x$-axis) and the depth of generated proof trees (indicated by color). When there are multiple correct proof trees, we plot the one with the maximum depth. As a comparison, we also plot the success rates of the proofs generated by the RL model trained w/o lemma proposal. Fig. 4 shows that the improvement of ProD-RL mostly comes from proving theorems with low-to-medium difficulty where the depth of the ground-truth proof is at most 2. For more complex theorems, both models' pass rates are low and the improvement of our method is not significant, meaning that they are currently beyond the models' capability.

### 4.4 CASE STUDY OF PROPOSED LEMMAS

In this section, we manually examine the new lemmas proposed during RL and list the typical cases where new lemmas are proposed. We emphasize that many AFP files focus on complex concepts

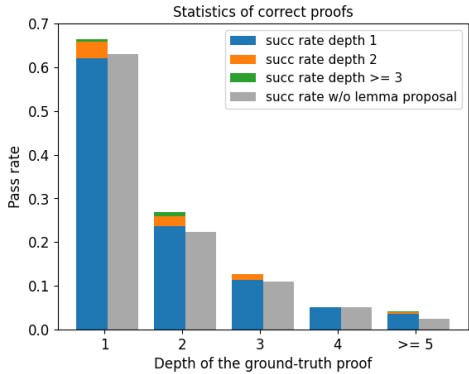

Figure 4: The pass rate of theorems in AFP test grouped by the depth of their ground-truth proof. Grey bars represent the proof generated by the model SFT w/o lemma proposal, and the colored bars represent the proof trees generated by ProD-RL with various depths.

and results in mathematics or computer science, making manual examination challenging. Therefore, examples in this section are biased toward easier theorems.

**Case 1: Model decomposes theorems into lemmas.** In this case, the model correctly decomposes the proof of a theorem into several lemmas. The following example belongs to the AFP file `List-Infinite`, which focuses on lists and sets with infinite size. The theorem (Line 1) states that the cardinality of the set $A \cup \{x\}$ equals $|A|$ if $x \in A$, or the successor integer of $|A|$ otherwise (i.e., $|A| + 1$ for finite $A$ and $\infty$ otherwise). During the proof (Lines 2-4), our model proposes two lemmas in Lines 2 and 3 to deal with the two possible cases ($x \notin A$ or $x \in A$) respectively. Finally, Line 4 proves the original theorem using the two proposed lemmas.

```
1  theorem icard_insert_if: "icard (insert x A) = (if x ∈ A then icard A else eSuc (icard A))"
2    <invoke> lemma icard_insert_disjoint: "x ∉ A ⟹ icard (insert x A) = eSuc (icard A)" </invoke>
3    <invoke> lemma icard_insert_eq: "x ∈ A ⟹ icard (insert x A) = icard A" </invoke>
4    by (simp add: icard_insert_eq icard_insert_disjoint)
```

**Case 2: The proposed lemma is a rephrase of an existing lemma.** We also find that some proposed lemmas are rephrases of existing lemmas in the training dataset. Although in this case the proposed lemma is not fundamentally useful for proving new theorems, they can be viewed as data augmentation to enhance the models' performance. In the following example, the model produces a lemma equivalent to one in an AFP file. Line 1 shows the original form of the lemma stated in the AFP file, while Lines 2-4 show an equivalent lemma proposed by our model during RL.

```
1  lemma icard_mono: "A ⊆ B ⟹ icard A ≤ icard B"
```

```
2  lemma icard_mono:
3    assumes "A ⊆ B"
4    shows "icard A ≤ icard B"
```

**Case 3: The proposed lemma is novel but not useful to the original proof.** We also observe cases where the proposed lemma is novel, but the conditional proof of the theorem is incorrect. In following example, the proposed lemma states that the shortest path between vertices $u, v$ is a lower bound for the length of any path that connects $u, v$ (in an unweighted and undirected graph):

```
1  lemma shortest_path_lower_bound:
2    assumes "p ∈ connecting_paths u v"
3    shows "shortest_path u v ≤ enat (walk_length p)"
```

This lemma is proposed to prove that the shortest path between the vertex $u$ and itself has length 0 (which is a theorem in the AFP file). However, the conditional proof of the theorem contains a few mistakes while the proposed lemma is proved separately. In this case, we still train on the correct lemma even though it might not be directly useful to the theorem in the training set.

**Remarks.** We observe that the lemmas proposed by the model typically do not involve complex ideas. We attribute this to two main factors: (a) the limited size of our model and formal proof dataset, and (b) the fact that many human-written lemmas in the AFP file are indeed about basic facts and basic properties (which are often used to prove more complex theorems later). Nevertheless, our model still proposes and proves reasonable lemmas that are not present in the training dataset, and our experiments demonstrate that with these proposed lemmas, ProD-RL outperforms ProD-SFT on holdout test sets. We leave it to future work to scale up our method and force the model to focus on more challenging theorems.

## 5 RELATED WORKS

To generate formal proofs with language models, most prior methods provide the verifier's state to the model to sample the proofs step-by-step, and use algorithms like MCTS to search for a correct complete proof (Han et al., 2021; Jiang et al., 2021; 2022b; Lample et al., 2022; Polu & Sutskever, 2020; Polu et al., 2022). The major drawback of these methods is their high computation cost at test time. Recent works (First et al., 2023) train a large language model to generate a whole proof directly without the verifier's state. Our baseline, the SFT model without lemma proposal, can be viewed as a reproduction of their methods with a slightly different way of computing the context.

Prior works also use reinforcement learning or expert iteration to improve the models' performance on writing formal proofs, where the training datasets contain formal synthetic inequalities (Polu et al., 2022) or statements translated from natural language mathematical problems (Wu et al., 2022; Xin et al., 2024a;a;b). In contrast, we aim to improve the models' performance without any additional (even unlabeled) data. As future works, we could run ProD-RL with additional datasets.

Another line of research aims to translate natural language proofs into formal proofs (Jiang et al., 2022a; Zheng et al., 2023). Xin et al. (2023) build a library of useful lemmas by decomposing natural language proofs into lemmas with an LLM and then formalizing the decomposed proofs. In contrast, we propose new lemmas entirely in formal language.

Automated theorem provers (ATPs) have been extensively studied, with various learning- or search-based methods developed to generate tactics for a given proof state (e.g., Gauthier et al. (2021); Schulz et al. (2019) and references therein). In comparison, our method focuses on generating multi-step proofs with LLMs while using existing ATP tools to complete low-level details. Orthogonally, a recent method (Mikuła et al., 2023) improves existing provers with a transformer-based retrieval model as the premise-selection tool, and could potentially be combined with our methods.

In general, mathematical question-answering tasks (such as GSM8K (Cobbe et al., 2021) and MATH (Hendrycks et al., 2021)) and theorem-proving tasks (such as Welleck et al. (2021)) are well-accepted benchmarks for the reasoning capability of large language models. Prior works show that instruction tuning or RL can significantly improve the models' performance (Shao et al., 2024). However, evaluation on these tasks is either performed by another language model (which is prone to errors) (Lightman et al., 2023), or requires ground-truth answers that are hard to acquire at scale.

## 6 CONCLUSION

In this paper, we design a reinforcement learning algorithm that encourages LLMs to write formal proofs by decomposing them hierarchically. We also design a more natural testing setup by removing the directly relevant lemmas from the context. We show that, by proposing and proving new lemmas that are not present in the training dataset, the resulting model ProD-RL outperforms baselines trained on the same dataset.

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

---

**Algorithm 2** Generate proof trees (train)

---

1: **Inputs:** Model $\pi_\theta$, theorems (represented by tuples of context, statement, and condition proof) $G = \{(c_i, s_i, \rho_i^\star)\}_i$, maximum depth $d$.
2: **for** $\iota \leftarrow 0, 1, \cdots, d$ **do**
3:     Compute the invoke probability $\forall i$, $p_i = \frac{\pi_\theta(\texttt{<use\_invoke>}|c_i, s_i)}{\pi_\theta(\texttt{<use\_invoke>}|c_i, s_i) + \pi_\theta(\texttt{<no\_invoke>}|c_i, s_i)}$.
4:     Let $\kappa$ be the 50% quantile of $\{p_i\}_i$.
5:     **if** $\iota < d$ **then**
6:         $\widehat{G} = \{(c_i, s_i, \rho_i^\star) \mid p_i \geq \kappa \text{ or } u_i < p_i \text{ where } u_i \sim \text{Unif}[0, 1]\}$.
7:     **else**
8:         $\widehat{G} = \emptyset$.
9:     Sample proofs $\rho_i \sim \pi_\theta(\cdot \mid c_i \, s_i \, \texttt{<no\_invoke>})$ for lemmas $(c_i, s_i, \rho_i^\star)$ in $G$.
10:    Sample proofs $\widehat{\rho}_i \sim \pi_\theta(\cdot \mid c_i \, s_i \, \texttt{<use\_invoke>})$ for lemmas $(c_i, s_i, \rho_i^\star)$ in $\widehat{G}$.
11:    $P_\iota \leftarrow \{(c_i, s_i, \rho_i) \mid \forall i \text{ s.t. } (c_i, s_i, \rho_i^\star) \in G\} \cup \{(c_i, s_i, \widehat{\rho}_i) \mid \forall i \text{ s.t. } (c_i, s_i, \rho_i^\star) \in \widehat{G}\}$.
12:    **if** $\iota = 0$ **then**
13:        $P_\iota \leftarrow P_\iota \cup G$.      ($\triangleright$) In training, we also complete proof trees for ground-truth proofs.
14:    Extract proposed lemmas (note that a condition proof $\rho$ might propose more than one lemma $l$): $G \leftarrow \{(c, l, \text{Null}) \mid (c, s, \bar{\rho}) \in P_\iota \text{ and } l \text{ is proposed in } \bar{\rho}\}$.
15: **Return** $\cup_{\iota=0}^d P_\iota$.

---

Huajian Xin, Haiming Wang, Chuanyang Zheng, Lin Li, Zhengying Liu, Qingxing Cao, Yinya Huang, Jing Xiong, Han Shi, Enze Xie, et al. Lego-prover: Neural theorem proving with growing libraries. *arXiv preprint arXiv:2310.00656*, 2023.

Huajian Xin, Daya Guo, Zhihong Shao, Zhizhou Ren, Qihao Zhu, Bo Liu, Chong Ruan, Wenda Li, and Xiaodan Liang. Deepseek-prover: Advancing theorem proving in llms through large-scale synthetic data. *arXiv preprint arXiv:2405.14333*, 2024a.

Huajian Xin, ZZ Ren, Junxiao Song, Zhihong Shao, Wanjia Zhao, Haocheng Wang, Bo Liu, Liyue Zhang, Xuan Lu, Qiushi Du, et al. Deepseek-prover-v1. 5: Harnessing proof assistant feedback for reinforcement learning and monte-carlo tree search. *arXiv preprint arXiv:2408.08152*, 2024b.

Chuanyang Zheng, Haiming Wang, Enze Xie, Zhengying Liu, Jiankai Sun, Huajian Xin, Jianhao Shen, Zhenguo Li, and Yu Li. Lyra: Orchestrating dual correction in automated theorem proving. *arXiv preprint arXiv:2309.15806*, 2023.

Kunhao Zheng, Jesse Michael Han, and Stanislas Polu. minif2f: a cross-system benchmark for formal olympiad-level mathematics. In *International Conference on Learning Representations*, 2021.

## A    ADDITIONAL EXPERIMENTS DETAILS

### A.1    GENERATING PROOF TREES FOR RL

In Alg. 2, we present the algorithm for generating proof trees during RL training. Recall that, compared with Alg. 1, there are two major differences:

(a) for the theorems where the probability $\pi_\theta(\texttt{<use\_invoke>} \mid x)$ is among the top 50% in the batch, we will force the model to generate conditional proofs with $t_0 = \texttt{<use\_invoke>}$ (Line 4-6), and

(b) for every theorem where the model generates a conditional proof with new lemmas, we also let the model generate another conditional proof without proposing new lemmas (Line 11).

### A.2    TRAINING DETAILS OF BASELINE MODELS

In this section, we describe the additional details for training the baseline models using reinforcement learning.

Our RL training pipeline for the baseline models is similar to that of ProD-RL, except that the models only generate proofs without lemma proposal. For RL baselines, we use the same dataset and the same

hyperparameters as our method. To mix the ground-truth conditional proofs with generated proofs, we convert the conditional proofs to proofs without lemma proposal by moving all the proposed lemma in the conditional proof to the context.

## A.3 ADDITIONAL EXPERIMENT DETAILS

**Details of using sledgehammer in the proof.** Sledgehammer is a premise selection tool that can automatically generate proofs to solve the current goal. Although sledgehammer is not always applicable, Jiang et al. (2022b) shows that letting the model to call sledgehammer whenever it is applicable greatly improves the model's performance.

To let the model use sledgehammer, we replace the actual proof steps in the training dataset by a call to sledgehammer if the proof step either (a) contains the proof tactics 'meson, metis, and smt' (these tactics are typically generated by sledgehammer), or (b) belongs to a predefined simple set of proof tactics that can be easily generated. In particular, they are

```
[by auto, by simp, by blast, by fastforce, by force,
by eval, by presburger, by sos, by arith, by linarith,
by (auto simp: field_simps)]
```

When testing a generated proof with calls to sledgehammer, we follow the pipeline of (Jiang et al., 2022a) — first, we try to replace the 'sledgehammer' command by one of the predefined tactics. If all the attempts fail, we call the actual premises selection tool in Isabelle with a 10s timeout. If the tool does not return a valid proof, we consider this step incorrect.

Note that Jiang et al. (2022b) decide when to replace the actual proof step by a call to sledgehammer more aggressively. They attempt to call sledgehammer at every proof step, and replace the actual proof step with sledgehammer if the attempt is successful. In contrast, our decision is made without interacting with the formal verifier. This is because applying sledgehammer to every proof step requires a lot of compute, which would significantly slow down the reinforcement learning process.

**Dataset split.** Here we describe how to split the training and test data based on the dependency of the AFP files. We first compute the dependency graph by crawling the AFP website `https://www.isa-afp.org/entries/`, which lists the dependency of the AFP entries. Then we find the set of AFP entries that all other entries do not depend on using the dependency graph, in which we randomly sample 10% of the entries as the holdout test set. The resulting holdout entries are:

```
[Verified_SAT_Based_AI_Planning, SIFPL, Khovanskii_Theorem,
Bondy, Rewriting_Z, Decreasing-Diagrams-II, Registers,
LocalLexing, FeatherweightJava, FFT, Knot_Theory, Eval_FO,
Saturation_Framework_Extensions, Hales_Jewett, SPARCv8,
CoSMeDis, LP_Duality, PAPP_Impossibility, Groebner_Macaulay,
Abstract-Hoare-Logics, PCF, Jordan_Hoelder, Knights_Tour,
FOL_Seq_Calc3, Cartan_FP, InformationFlowSlicing_Inter, LOFT,
Diophantine_Eqns_Lin_Hom, Dynamic_Tables, Schutz_Spacetime,
Elliptic_Curves_Group_Law, ArrowImpossibilityGS,
Goodstein_Lambda, XML, GenClock, Topological_Semantics].
```

**Additional training detail.** We use the Llemma code base (`https://github.com/EleutherAI/math-lm`) for finetuning and updating the model in reinforcement learning. The discount factor used to compute the weight is $\gamma = \exp(-0.0005)$.

## A.4 COMPUTE RESOURCES

For supervised finetuning and reinforcement learning, we use a machine with 8 A100-80G GPUs. It takes approximately 8 GPU days in total (i.e., 1 day wall-clock time on a single machine with 8 GPUs) to finetune a 7B model on 300k examples for 2 epochs. It takes approximately 30 GPU hours to run a single RL experiment.

To generate proofs using the trained model, we use a mix of A100-80G and A5000 GPUs. On 8 A5000 GPUs, generating proof trees of depth 2 for 4k test examples takes about 1-2 hours, depending on the length of the proof and the number of proposed lemmas.

### A.5 LICENSES FOR EXISTING ASSETS

In this section, we list the licenses for existing assets used in this paper.

- LLemma (Azerbayev et al., 2023): Llama 2 Community License Agreement
- Archive of Formal Proofs: GNU LGPL
- Portal to ISAbelle (Jiang et al., 2021): BSD 3-Clause License
- Isabelle (Nipkow et al., 2002): BSD licenses
- miniF2F (Zheng et al., 2021): MIT License

## B ADDITIONAL RESULTS

In this section, we present additional experimental results.

### B.1 THE EFFECT OF SAMPLING TEMPERATURES

In our preliminary experiments, We tune the sampling temperature using the models trained on the AFP training sets $D_{\text{train}}^{\text{w/l}}$ and $D_{\text{train}}^{\text{w/o l}}$ (that is, training sets constructed with and without helper lemmas, respectively). We test the model on the AFP test set $D_{\text{test}}^{\text{w/o l}}$ with different temperatures to decide the best choice for testing our models. Fig. 5 shows the performance of the SFT model without lemma proposal using different sampling temperatures. We conclude that the temperature 0.7 is best for testing both models.

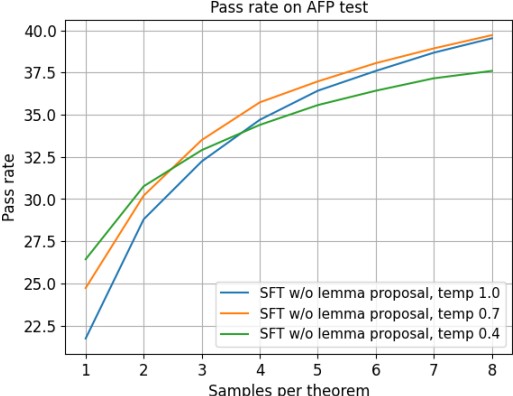

Figure 5: Pass rate of the SFT model without lemma proposal tested with different sampling temperature. We observe that lower temperature leads to better performance with 1 sample per theorem, and mildly larger temperature have better performance with more samples.

### B.2 VARYING THE SIZE OF THE RL DATASET

In this section we report the performance of the ProD-RL models trained with different rounds. Recall that in each round we generate proofs to a batch of 5k examples. Therefore equivalently, Figure 6 shows the performance of ProD-RL with a smaller RL dataset. Note that although the performance is not monotone with respect to the RL dataset size (e.g., ProD-RL at round 15 is worse than ProD-RL at round 10), which might due to training instability, all the RL models significantly outperforms the baseline ProD-SFT.

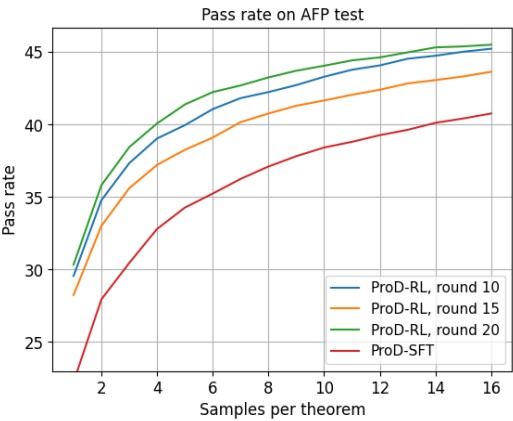

Figure 6: Pass rate of models trained with different rounds of RL training. Recall that in each round we generate proofs to a batch of 5k examples. The model trained with 20 rounds of RL achieves the best performance, and all RL models outperforms the baseline ProD-SFT model.

