# OpenReview forum: "Formal Theorem Proving by Rewarding LLMs to Decompose Proofs Hierarchically"
_ICLR.cc/2025/Conference — Submitted to ICLR 2025_

### Official Review · Reviewer_vahD · 2024-10-24

**Soundness:** 2
**Presentation:** 3
**Contribution:** 1
**Rating:** 3
**Confidence:** 4

**Summary:**

This paper explores neural theorem proving in a setting where existing lemmas cannot be used and proposes a RL-based approach that encourages the model to decompose proofs into multiple subgoals and propose new lemmas to prove. The authors demonstrate that their approach, which combines supervised fine-tuning with reinforcement learning, achieves superior performance on the AFP dataset compared to baseline methods, including those that do not propose new lemmas or rely solely on supervised fine-tuning.

**Strengths:**

* The paper is well-written, clearly structured, and easy to follow.

* The proposed method is well-motivated. Experiments on the AFP dataset effectively demonstrate the advantages of lemma proposal with reinforcement learning.

**Weaknesses:**

* The idea of decomposing proofs into subgoals and proposing new lemmas is not entirely novel, as it has been explored in previous work. For instance, LEGO-Prover [1] uses informal proofs to break down proofs into manageable subgoals and leverages language models to either propose new lemmas or retrieve existing ones from a growing library. While the proposed setting of excluding relevant premises is indeed challenging, I believe it does not always reflect practical scenarios. Incorporating both lemma selection and lemma proposal would present a more generalized and realistic approach.

* The experiments are conducted exclusively on the AFP dataset, which may not provide a thorough evaluation of the method's robustness, particularly in out-of-distribution scenarios like the miniF2F dataset, a more commonly used benchmark. As a result, it is unclear how well the proposed approach generalizes beyond the training set. Additionally, in Case 2, there seems to be evidence that the model may be memorizing lemmas from the training data rather than proposing genuinely novel ones, which raises concerns about its ability to generate useful new lemmas in unseen contexts.

* Minor Points: There are some missing references to works that share similar ideas or components with this paper [2,3,4]. For instance, POETRY [4] introduces a recursive approach that decomposes proofs into subgoals and generates lemmas in a hierarchical, level-by-level manner. A broader overview of related work can be found in [5]. Additionally, the statement in Line 80 ("the best method along this line of research requires more than 1k GPU days with A100s to train a model with 600M parameters") may not be entirely accurate. More recent methods, such as InternLM-Step Prover [6], appear to have surpassed HTPS in both efficiency and requiring less computational resources.

[1] LEGO-Prover: Neural Theorem Proving with Growing Libraries

[2] TacticZero: Learning to Prove Theorems from Scratch with Deep Reinforcement Learning

[3] Proving Theorems Using Incremental Learning and Hindsight Experience Replay

[4] Proving Theorems Recursively

[5] A Survey on Deep Learning for Theorem Proving

[6] LEAN-GitHub: Compiling GitHub LEAN Repositories for a Versatile LEAN Prover

**Questions:**

* Could you evaluate the proposed method on the miniF2F benchmark? This dataset is more commonly used and would help assess the generalization of your approach beyond AFP.

* The paragraph starting at Line 270 suggests that the reward in the RL setting is based on the conditional proof, with its weight influenced by the conditional proof as well. However, in practice (as described in Line 277), it appears that the proposed method uses a language model as a value network to predict True/False, and the probability is used as the reward score, without incorporating any information from the conditional proof. Could you clarify why the conditional proof is discarded for the value function? Additionally, in the RL w/o lemma proposal, what rewards or weights are used in the absence of lemmas?

* Why does RL w/o lemma proposal underperform SFT w/o lemma proposal? For ProD-RL, you first apply SFT, followed by RL, which (almost) ensures that ProD-RL always outperforms ProD-SFT. Does RL w/o lemma proposal also involve an initial SFT stage? If so, it seems that RL w/o lemma proposal negatively impacts performance, and I would like more details on why this happens (since there are no details provided). If not, the comparison between RL w/o lemma proposal and ProD-RL may not be entirely fair.

---

> ### Author Response · Authors · 2024-11-19
> **Thank you for the comments (1/2)**
>
> We thank Reviewer vahD for their comments. In the following, we address the reviewer’s question in detail.
>
> > While the proposed setting of excluding relevant premises is indeed challenging, I believe it does not always reflect practical scenarios. Incorporating both lemma selection and lemma proposal would present a more generalized and realistic approach.
>
> We thank the reviewer for the constructive comments. In fact, combining our methods with premise selection tools such as MagnusHammer is straightforward, as we can simply replace Sledgehammer with MagnusHammer. Since we cannot find an open-sourced implementation of MagnusHammer, we leave this direction as future work.
>
> > In Case 2, there seems to be evidence that the model may be memorizing lemmas from the training data rather than proposing genuinely novel ones, which raises concerns about its ability to generate useful new lemmas in unseen contexts.
>
> We acknowledge that the model may memorize lemmas in some cases. However, in both Cases 1 and 3, the lemma proposed by our model is indeed useful since the proposed lemmas are not present in the training dataset. Figure 4 also demonstrates that, with lemma proposals, our ProD-RL model outperforms baseline methods by proposing lemmas in the test dataset where the context of theorems is never seen by the model during training.
>
> > There are some missing references to works that share similar ideas or components with this paper…
>
> We thank the reviewer for the additional references. We will update our paper accordingly upon revision.
>
> > The experiments are conducted exclusively on the AFP dataset, which may not provide a thorough evaluation of the method's robustness, particularly in out-of-distribution scenarios like the miniF2F dataset, a more commonly used benchmark. … Could you evaluate the proposed method on the miniF2F benchmark?
>
> On miniF2F, we find that our method does not have significant improvement over the baselines. We hypothesize the reasons to be: (a) the theorems are very different from theorems in the training dataset, and (b) we also observe that when tested on miniF2F theorems, our model failed to propose meaningful lemmas. This may be because proving to miniF2F-level mathematics questions typically does not require hierarchical decomposition. In contrast, the AFP datasets work with abstract mathematical objects and theorems, which intrinsically have hierarchical decompositions.
>
> Nonetheless, our results on miniF2F are shown in the following table:
>
> |               | SFT w/o lemma proposal | RL w/o lemma proposal | ProD-SFT | ProD-RL |
> |:-------------:|:----------------------:|:---------------------:|:--------:|:-------:|
> | miniF2F valid |          46.3          |          40.6         |   44.7   |   41.4  |
> |  miniF2F test |          40.6          |          38.9         |   39.3   |   39.3  |
>
> >  The paragraph starting at Line 270 suggests that the reward in the RL setting is based on the conditional proof, with its weight influenced by the conditional proof as well. However, in practice (as described in Line 277), it appears that the proposed method uses a language model as a value network to predict True/False, and the probability is used as the reward score, without incorporating any information from the conditional proof. Could you clarify why the conditional proof is discarded for the value function?
>
> We do not discard the conditional proof when computing the reward. For a theorem with conditional proof `p`, we only use the value function to compute the probability that the current model can prove the proposed lemmas.
>
> As a concrete example, suppose the conditional proof to the theorem proposes two lemmas `l1, l2` and the conditional proof itself is locally correct (meaning that it proves the theorem using lemmas `l1` and `l2`, assuming both lemmas are correct). Then the reward of this conditional proof is `V(l1) * V(l2)` times the additional discount factor.
>
> In comparison, we can also simply set a binary reward whose value depends on whether the proofs to lemmas `l1` and `l2` are also correct. If the value function is perfect, the two ways of computing the reward have the same expected value, and using the value function is a common trick to reduce the variance of the reward estimation.
>
> >  Additionally, in the RL w/o lemma proposal, what rewards or weights are used in the absence of lemmas?
>
> Without lemma proposal, the reward of a conditional proof is simply the correctness of the proof.

---

> > ### Author Response · Authors · 2024-11-19
> > **Thank you for the comments (2/2)**
> >
> > > Why does RL w/o lemma proposal underperform SFT w/o lemma proposal? For ProD-RL, you first apply SFT, followed by RL, which (almost) ensures that ProD-RL always outperforms ProD-SFT. Does RL w/o lemma proposal also involve an initial SFT stage? If so, it seems that RL w/o lemma proposal negatively impacts performance, and I would like more details on why this happens (since there are no details provided). If not, the comparison between RL w/o lemma proposal and ProD-RL may not be entirely fair.
> >
> > Yes, the RL w/o lemma proposal involves an initial SFT stage. In our MDP setup, the state is the theorem’s statement and the action is the whole proof. Therefore, running RL in our setup without lemma proposal cannot generate data with new MDP states. In other words, the naive RL algorithm on top of the SFT model in our setup cannot collect fundamentally new data, since the ground-truth action for every MDP state in the dataset is already given in the SFT dataset. Further RL training may simply cause overfitting problems. In contrast, with lemma proposals, we can generate novel data MDP states (i.e., the lemma statements) and therefore boost the performance of the SFT model.
> >
> > To reconcile prior works that show naive RL can further improve the SFT model, such as GPT-f [1] and PACT [2], we note that in GPT-f and PACT, the state is the verifier’s internal proof state (plus some contexts) and the action is a single proof step (and they rely on an additional search algorithm to find a correct multi-step proof). Therefore, running RL in GPT-f’s setup can generate data with new MDP states because different proofs may lead to different verifier’s proof states, and therefore enriches the dataset.
> >
> > We hope that our response addressed the reviewer’s concerns, and we are happy to answer any follow-up questions.
> >
> > [1] Polu, Stanislas, and Ilya Sutskever. "Generative language modeling for automated theorem proving." arXiv preprint arXiv:2009.03393 (2020).
> >
> > [2] Han, Jesse Michael, et al. "Proof artifact co-training for theorem proving with language models." arXiv preprint arXiv:2102.06203 (2021).

---

> ### Comment · Reviewer_vahD · 2024-11-21
>
> Thank you for your detailed response. I have a few follow-up questions and suggestions for clarification and improvements to the paper.
>
> 1. Regarding the Use of Sledgehammer
> - Has Sledgehammer been used into the proposed approach? Specifically, is it used to fill proof gaps for the proposed lemmas?
> - If Sledgehammer is indeed part of the method, I recommend providing additional details about its integration, as the current description includes very limited discussion. Furthermore, rather than independently proposing every lemma, have you considered leveraging existing lemma retrieval techniques like those in [1]? This approach could potentially address a broader range of cases and enhance the method's generality.
> - If Sledgehammer is not used, and proof steps are still predicted by LLMs, why not consider modeling each proof step as an individual action in the RL framework? This would better align with existing RL approaches to theorem proving.
>
> [1]  LEGO-Prover: Neural Theorem Proving with Growing Libraries
>
> 2. On the RL Formulation
>
> If the action in your RL w/o lemma proposal setup is defined as generating the entire proof, this reduces the RL problem to a bandit-like setting, as there is no exploration during the proving process. This formulation faces significant challenges:
> - The action space is almost infinite.
> - Correct proofs often involve lengthy token sequences but rewards are extremely sparse.
>
> These challenges likely contribute to the observed underperformance of RL w/o lemma proposal compared to SFT w/o lemma proposal. The lack of intermediate structure or guidance in this setup may also have a negative impact on the learning process. I suggest adding a more detailed settings of the baseline methods and discussing these challenges explicitly to provide further clarity.
>
> 3. Performance on miniF2F
>
> - You mention that your method does not show significant improvement (and indeed has a negative impact) over baselines on miniF2F, potentially due to the lack of hierarchical decomposition in these problems. However, I believe miniF2F does exhibit hierarchical decomposition (albeit different from AFP), which is why methods like LEGO-Prover perform well in this domain.
> - I posit that the underperformance of your approach may be more related to its generalization ability. Including these results in your paper to clearly illustrate the limitations of your method would be beneficial and add transparency.
>
> 4. Insights on Novelty and Contributions
>
> - From my understanding, decomposing theorems into lemmas is intuitive for declarative proof assistants like Isabelle and has been extensively explored in the literature. The main contribution of your work appears to be the use of RL to train a lemma proposal mechanism. Beyond this, can the authors provide additional insights into the novelty or other valuable aspects of the proposed approach?
>
> - One potential reason for your improved performance on AFP could be that RL facilitates the generation of more diverse intermediate lemmas compared to purely SFT approaches. This diversity could better guide the proving process. I suggest analyzing and comparing the diversity of lemmas generated by ProD-RL and SFT without lemma proposal to evaluate this potential advantage.

---

> > ### Author Response · Authors · 2024-11-21
> > **Thank you for your additional comments (1/2)**
> >
> > We thank reviewer vahD for their additional constructive comments and suggestions. In the following, we address the reviewer’s comments in detail.
> >
> > > Has Sledgehammer been used into the proposed approach? Specifically, is it used to fill proof gaps for the proposed lemmas? If Sledgehammer is indeed part of the method, I recommend providing additional details about its integration.
> >
> > Yes, Sledgehammer is used in our proof for both the original theorem and the proposed lemmas. For both the theorems and the proposed lemmas, we use the language model to generate a complete proof that may have sledgehammer commands in the middle of the proof.
> >
> > To use Sledgehammer, we mostly follow the methodology in Thor [1] — we first augment the proofs in the SFT dataset to replace tactics like `meson`, `smt`, and others like `by simp` by a single word `Sledgehammer` (intuitively, these are the tactics that can be found by Sledgehammer), and then the model can directly generate a whole-proof with Sledgehammer commands in the middle. When testing the correctness of a proof, we simply call the Sledgehammer tool to complete a single-step proof. Unlike Thor, we don’t interactively call the LLM and the Isabelle prover because we generate a whole proof once and then test its correctness.
> >
> > We will update the paper accordingly upon revision.
> >
> > [1] Jiang, Albert Qiaochu, et al. Thor: Wielding hammers to integrate language models and automated theorem provers."
> >
> > > Furthermore, rather than independently proposing every lemma, have you considered leveraging existing lemma retrieval techniques like those in [1]? This approach could potentially address a broader range of cases and enhance the method's generality.
> >
> > We thank the reviewer for the suggestion! Yes, this direction is something we are actively considering. We believe that retrieval techniques like LEGO-Prover or MagnusHammer can indeed enhance the model’s performance and even boost its generalization. Our framework is naturally compatible with retrieval techniques since we can either use RAG by prepending the retrieved lemmas in the context, or directly replace Sledgehammer by MagnusHammer. For this particular paper, however, we decided to leave this direction for future work since our contribution is mostly orthogonal.
> >
> > > […] why not consider modeling each proof step as an individual action in the RL framework? This would better align with existing RL approaches to theorem proving.
> >
> > We choose to use REINFORCE instead of RL algorithms mostly for the conceptual and implementation simplicity. In addition, we’d also like to explore the potential of whole-proof generation instead of search + single step generation because of its efficiency and compatibility to common LLM infrastructures. Incorporating the step-level feedback from Isabelle in RL training is an interesting direction for future works.
> >
> > > If the action in your RL w/o lemma proposal setup is defined as generating the entire proof, this reduces the RL problem to a bandit-like setting, as there is no exploration during the proving process. [...] These challenges likely contribute to the observed underperformance of RL w/o lemma proposal compared to SFT w/o lemma proposal. The lack of intermediate structure or guidance in this setup may also have a negative impact on the learning process.
> >
> > We agree with the reviewer that it essentially reduces the RL problem to a bandit-like setting. The resulting algorithm is actually almost equivalent to expert iteration in the recent literature. In fact, even for ProD-RL, we use REINFORCE to update the policy, which is also almost equivalent to expert iteration (except that we use a different way to assign the reward). Therefore, we generally don’t think the training algorithm is the key factor that causes the difference between RL w/o lemma proposal and *ProD-RL*. However, it is indeed possible that, by using the intermediate feedbacks, both RL w/o lemma proposal and Prod-RL could have a better performance, and RL w/o lemma proposal could outperforms SFT w/o lemma proposal.
> >
> > We will add a detailed discussion on the performance of RL w/o lemma proposal upon revision.
> >
> > > However, I believe miniF2F does exhibit hierarchical decomposition (albeit different from AFP), which is why methods like LEGO-Prover perform well in this domain. I posit that the underperformance of your approach may be more related to its generalization ability. Including these results in your paper to clearly illustrate the limitations of your method would be beneficial and add transparency.
> >
> > Thank you for the comments! Yes, we agree that the generalization ability might be a key reason to explain ProD-RL’s performance on miniF2F. We will include these results in the final version of our paper accordingly.

---

> > > ### Author Response · Authors · 2024-11-21
> > > **Thank you for your additional comments (2/2)**
> > >
> > > > From my understanding, decomposing theorems into lemmas is intuitive for declarative proof assistants like Isabelle and has been extensively explored in the literature. The main contribution of your work appears to be the use of RL to train a lemma proposal mechanism. Beyond this, can the authors provide additional insights into the novelty or other valuable aspects of the proposed approach?
> > >
> > > In addition to the RL training pipeline, our contributions are:
> > >
> > > - Our reward design can be seen as an instantiation of the Hindsight Experience Replay (HER) algorithm, which is specifically designed for sparse reward RL problems. In particular, even if the theorem is not proved, we can still train on the correct proofs of proposed lemmas. This algorithm somewhat imitates a mathematician’s process when facing an extremely challenging open question, though our model is trained on simpler questions due to its limited size.
> > > - Although the idea of using HER with formal proofs is explored before [2], it was limited to synthetic questions where hindsight trajectories can be generated easily. Our method provides a more elegant/natural way of generating the hindsight trajectories.
> > > - To the best of our knowledge (please kindly let us know of any missing related works in this direction otherwise), prior works like LEGO-Prover require a strong model such as GPT4 to decompose the theorems into lemmas, and its reasoning steps are performed in natural language so that it’s closer to the pretraining dataset. In contrast, our method designs a framework of hierarchical decomposition purely in formal language, and is not restricted to synthetic statements with good proof structures.
> > >
> > > [2] Aygün, Eser, et al. "Proving theorems using incremental learning and hindsight experience replay." International Conference on Machine Learning. PMLR, 2022.
> > >
> > > > One potential reason for your improved performance on AFP could be that RL facilitates the generation of more diverse intermediate lemmas compared to purely SFT approaches. This diversity could better guide the proving process. I suggest analyzing and comparing the diversity of lemmas generated by ProD-RL and SFT without lemma proposal to evaluate this potential advantage.
> > >
> > > Thank you for the comments! We indeed find that our RL algorithm generates 37% more novel lemmas after deduplication with exact match and the Isabelle built-in `solve_direct` method. We also agree with the reviewer that this is one of the main reasons for its improvement over SFT. In general, accessing the diversity of generated lemmas on the semantic level is not an easy task because it is hard to determine the similarity between lemmas. The best way we can think of right now is the case study in Section 4.4, plus some simple statistics with exact match deduplication.
> > >
> > > A potential evaluation method is to look at the cosine similarity of embeddings of the lemmas, but then the quality of the analysis highly depends on the embedding model itself, which is less ideal. Please kindly let us know if there are better ways to approach this question.
> > >
> > > Finally, we thank reviewer vahD again for this constructive discussion, and we are happy to answer any follow-up questions.

---

> ### Comment · Reviewer_vahD · 2024-11-26
>
> Thank you for your response. I appreciate the detailed discussion and insights provided by the authors. However, I believe the paper, in its current form, still requires significant improvement in clarity and a more comprehensive analysis and refinement of the experimental results, to be considered for acceptance. While the proposed method is intuitive and well-motivated, I could hardly find it "novel" as the core ideas (decomposing theorems into lemmas, Hindsight Experience Replay) appear to overlap with existing work. Additionally, the authors emphasize that their approach operates entirely in formal language, in contrast to some related methods that leverage natural language. However, if leveraging natural language can yield better results, as demonstrated in [1], why not explore its use as a means to guide formal proofs to achieve better performance? Another critical concern is the generalization ability of the proposed method. The authors acknowledge that their model struggles to propose meaningful lemmas for problems in the miniF2F dataset and even underperforms basic SFT approaches. This suggests that even simple SFT exhibits better generalization than the proposed method. Without presenting a strategy to address this critical limitation or additional experiments demonstrating potential improvements in generalization, simply labeling it as a limitation feels insufficient for acceptance. In my opinion, achieving the best performance is not always necessary for a method to be considered impactful. However, underperforming compared to basic SFT makes it difficult to view this work as a "significant contribution" to the field in its current state.
>
> [1] Lean-STaR: Learning to Interleave Thinking and Proving

---

> > ### Author Response · Authors · 2024-11-27
> >
> > We thank the reviewer for the additional comments.
> >
> > First, we would like to point out that, in our opinion, there is a fundamental difference between this paper and the research that leverages powerful external LLMs. That is, where the training signal comes from.
> >
> > - First, since the miniF2F benchmark is created by translating natural language math questions into formal proofs, and existing LLMs are extensively trained on natural language data, it is not a surprise that leveraging natural language can improve the models’ performance on miniF2F. Therefore, if the goal is to advance the performance on miniF2F, leveraging natural language data/reasoning is a strong method.
> > - However, with this approach, it is unclear whether the improvement comes from the adaptation of the knowledge (that the models already have) to formal languages, or the improvement of their mathematical and reasoning capability. The research question in the former case is, while of interest on its own, orthogonal to this paper. There is no doubt that finetuning/prompt-engineering GPT4 is a way to improve the state-of-the-art performance on miniF2F.
> > - In contrast, this paper aims to explore the possibility of the latter case, where the training signal does not rely on any powerful teacher model and solely comes from the formal verifier like Isabelle. This direction resonates with the weak-to-strong generalization question, where the weak training signal is the correctness of the proofs which can be determined in polynomial time, and the strong capability we are aiming to achieve is generating correct proofs, which is even harder than NP-hard.
> > - Therefore we explicitly choose to not use any strong LLMs to perform reasoning steps in natural language. We carefully curated the AFP test datasets to make sure that the improvement of the performance is independent of potential knowledge leakage. On AFP datasets, our methods show significant improvement over baseline methods. MiniF2F, on the other hand, is not the major focus of this paper.
> >
> > Regarding the novelty of this paper, we kindly ask the reviewer to clarify the comments:
> > >  I could hardly find it "novel" as the core ideas (decomposing theorems into lemmas, Hindsight Experience Replay) appear to overlap with existing work.
> >
> > Does the reviewer mean that using Hindsight Experience Replay in the field of formal proofs overlaps with existing work? To the best of our knowledge, [2] is the closest related work in this field, which only works for first-order logics that have a well-structured format. Similarly, [1] decomposes theorems into lemmas by ChatGPT, but no training is involved; [3] trains a new model to search proofs hierarchically without HER.
> >
> > [1] Wang, Haiming, et al. Lego-prover: Neural theorem proving with growing libraries.
> >
> > [2] Aygün, Eser, et al. Proving theorems using incremental learning and hindsight experience replay.
> >
> > [3] Wang, Haiming, et al. Proving Theorems Recursively.

---

> ### Comment · Reviewer_vahD · 2024-11-27
>
> I appreciate the authors' quick response and would like to address my concerns in detail.
>
> - Regarding the novelty
>
> While I understand that assessing novelty can be subjective, I find the core idea of the proposed method overlaps with existing works. As you acknowledge, decomposing theorems into lemmas and using Hindsight Experience Replay are established techniques *within theorem proving research*, even if applied to different settings (e.g., alternative logics or LLMs). However, combining previously known approaches in a somewhat distinct setup does not, in my view, qualify as entirely ''novel''.
>
> - Regarding the generalization ability
>
> I appreciate the motivation to rely solely on formal verifiers like Isabelle without the assistance of powerful teacher models (i.e., "weak-to-strong generalization"). However, I find it unconvincing to downplay the limitations in generalization by stating that MiniF2F is not a major focus, especially considering this dataset is the most widely used benchmark in this field. Furthermore, the proposed method underperforms even simple SFT on the formal proof of the AFP dataset when testing on the MiniF2F. If the goal is to improve the reasoning abilities of weak LLMs, performance on problems beyond similar training instances is also crucial. This concern is amplified by the fact that problems in MiniF2F, which consist of high-school competition-level math, may be less challenging than those in the AFP dataset, which spans diverse mathematical research topics.
>
> In my view, while proposing entirely novel approaches is very commendable but not always essential, addressing generalization remains a critical factor. If the authors can present strategies to overcome these limitations and provide additional experiments showing potential improvements in generalization (better than simple SFT), I would like to reassess and raise my score.

---

> > ### Author Response · Authors · 2024-12-02
> >
> > We thank the reviewer for their clarification and the additional comments. Regarding the performance on miniF2F, we conducted the following experiments.
> >
> > If we focus on the performance on miniF2F, and given the fact that the questions in miniF2F are sufficiently different from AFP, we suggest that it is better to use our RL training pipeline as a data augmentation procedure. Specifically, we collect all the new lemmas proposed and proved by our model during RL, and combine them with the original SFT dataset (in other words, we discard the ProD-RL model but keep the generated data). Then, we train a new model with the combined dataset with the same hyperparameter as ProD-SFT. We call this new model ProD-SFT-aug.
> >
> > Note that ProD-SFT-aug will unavoidably underperform ProD-RL on in-domain test sets AFP test and AFP 2023. This is because the ProD-RL model is trained in an on-policy manner using RL. However, without the RL training, ProD-SFT-aug generalizes better to out-of-domain test sets miniF2F, as shown in the following table.
> >
> > |                        | miniF2F valid | miniF2F test |
> > |:----------------------:|:-------------:|:------------:|
> > | SFT w/o lemma proposal |      46.3     |     40.6     |
> > |        ProD-SFT        |      44.7     |     39.3     |
> > |         ProD-RL        |      41.4     |     39.3     |
> > |      ProD-SFT-aug      |      45.1     |     41.4     |
> >
> > On both miniF2F valid and miniF2F test, ProD-SFT-aug improves ProD-SFT, showing that the additional lemmas proposed by our model is indeed useful. We’d also like to note that the comparison between ProD-SFT-aug and the baseline model SFT w/o lemma proposal shows mixed signals. On the one hand, proposing correct lemmas itself is challenging, which may distract the ProD-SFT-aug model from learning to generate direct proofs. On the other hand, ProD-SFT-aug benefits from the additional training data generated by proposing and proving new lemmas.

---

### Official Review · Reviewer_CALz · 2024-11-01

**Soundness:** 3
**Presentation:** 2
**Contribution:** 2
**Rating:** 3
**Confidence:** 4

**Summary:**

This paper introduces ProD-RL, an RL method to enhance LLM’s theorem-proving capabilities by encouraging them to decompose proofs into lemmas. It avoids oversimplifying the task by removing direct lemma support from the training setup, requiring the model to independently propose intermediate steps in the proof process. The paper evaluates the model’s performance on the Isabelle AFP dataset and shows 45.5%/39.5% accuracy on AFP-test/AFP-2023, which outperforms SFT baselines.

**Strengths:**

1. This work proposes an RL-driven approach to proof decomposition without lemma reliance, an original contribution that aligns with similar ideas in [3], [4]
2. ProD-RL’s methodology yields improvements over SFT, highlighting the model’s capability to generalize beyond pre-existing lemmas, and trains on a more realistic dataset, splitting by dependency rather than at random.
3. The work contributes to LLM for theorem proving, an important direction for mathematical reasoning research.

**Weaknesses:**

1. Clarity and Readability: The paper is a little bit difficult to follow, especially in algorithmic explanations and conditional proof notation. The baseline and proposed methods are not clearly differentiated in the results. Improved organization, clearer definitions, and visuals would make the methodology more accessible.
2.  Generalization and Complexity: There is limited exploration of scalability to higher-complexity proofs. It's better to address whether ProD-RL is viable for deeper proof trees.
3.  Comparative Benchmarks and Dataset Exclusion: The absence of benchmarks like minif2f in experiments raises questions, given its mention in the license section. Also, there could be more comparisons with other theorem-proving techniques in the experiment section.

**Questions:**

1. Could the authors address the large experimental difference between ProD-RL (45.5%) and MagnusHammer’s 71% on the PISA benchmark [2]? A direct comparison/discussion of scalability/ablation study could strengthen the work.

2. Why is “MiniF2F” only in the licensing section without experiment inclusion? This benchmark is a robust test for theorem-proving tasks.
it could be better to add the evaluation PutnamBench [1] which assesses the methods on hard competition level problems.



[1] Tsoukalas, George, et al. "PutnamBench: Evaluating Neural Theorem-Provers on the Putnam Mathematical Competition." arXiv preprint arXiv:2407.11214 (2024).
[2] Mikuła, Maciej, et al. "Magnushammer: A transformer-based approach to premise selection." arXiv preprint arXiv:2303.04488 (2023).
[3] Wang, Haiming, et al. "Lego-prover: Neural theorem proving with growing libraries." arXiv preprint arXiv:2310.00656 (2023).
[4] Aygün, Eser, et al. "Proving theorems using incremental learning and hindsight experience replay." International Conference on Machine Learning. PMLR, 2022.

---

> ### Author Response · Authors · 2024-11-19
> **Thank you for the comments.**
>
> We thank Reviewer CALz for their comments. In the following, we address the reviewer’s question in detail.
>
> > Could the authors address the large experimental difference between ProD-RL (45.5%) and MagnusHammer’s 71% on the PISA benchmark [2]?
>
> First, there are two major differences between ours and PISA’s evaluation setup:
>
> - We split the test set based on the dependency of the files, while PISA splits the test theorems randomly. As a result, for the theorems in the PISA test set, there may exist proofs of similar theorems in the PISA training set.
> - When testing on a theorem, we do not allow the proof to directly use lemmas in the same AFP file. Instead, we let the model propose and prove every lemma it needs in the proof. In comparison, MagnusHammer focuses on training a retrieval model that selects useful lemmas from the context, including those in the same AFP file.
>
> As a result, our test setup is fundamentally harder than PISA. As shown in Table 1, the *same* model has much worse performance on our setup.
>
> Second, we also test the pass@64 performance of our baseline model on $D_{val}^{w/l}$, our reproduction of the PISA test set. The baseline model achieves 63.9% on the test set with Sledgehammer. In comparison, Thor with MagnusHammer archives 71%. We’d also like to emphasize that, according to First et al., (2023), whole-proof generation with 64 examples is 6x faster than search methods like Thor.
>
> Finally, the contribution of MagnusHammer is orthogonal to ours. In fact, combining our methods with MagnusHammer is straightforward, as we can simply replace Sledgehammer with MagnusHammer. We leave this direction as future work.
>
> > Why is “MiniF2F” only in the licensing section without experiment inclusion? This benchmark is a robust test for theorem-proving tasks. it could be better to add the evaluation PutnamBench [1] which assesses the methods on hard competition level problems.
>
> We use MiniF2F benchmark in the early development of this work. We find that our method does not have significant improvement over the baselines. We hypothesize the reasons to be: (a) the theorems are very different from theorems in the training dataset, and (b) we also observe that when tested on miniF2F theorems, our model failed to propose meaningful lemmas. This may be because proving to miniF2F-level mathematics questions typically does not require hierarchical decomposition. In contrast, the AFP datasets work with abstract mathematical objects and theorems, which intrinsically have hierarchical decompositions.
>
> > There is limited exploration of scalability to higher-complexity proofs. It's better to address whether ProD-RL is viable for deeper proof trees.
>
> We thank the reviewer for the suggestion. We observe that in AFP test sets, theorems that require deeper proof trees are too challenging for our current model since the pass@64 performance is below 10%, as shown in Figure 4. It might require scaling up the model or the dataset before we can test the scalability of our methods. Unfortunately, we are currently limited by the compute budget and therefore cannot perform a larger scale experiment.
>
> We hope that our response addressed the reviewer’s concerns, and we are happy to answer any follow-up questions.

---

### Official Review · Reviewer_CLmw · 2024-11-02

**Soundness:** 4
**Presentation:** 3
**Contribution:** 3
**Rating:** 8
**Confidence:** 4

**Summary:**

The paper proposes a novel paradigm in neural theorem proving. Previous methods assume pre-provided lemmas during the proving stage. In contrast, this paper encourages the model to decompose the theorem into lemmas, prove the lemmas, and then use these lemmas to prove the theorem. This approach more closely resembles a real-world theorem-proving process. And the effectiveness of the proposed framework is demonstrated by the experiments.

**Strengths:**

- The motivation for this work is strong, addressing a critical problem in the domain of neural theorem proving. The proposed framework is highly useful for this task.
- The experimental results demonstrate its effectiveness in creating lemmas during the theorem-proving process.
- The paper is well-written and easy to follow.

**Weaknesses:**

- The primary concern with the proposed method is its performance. While effective, the enhancement offered by the framework is marginal when compared to the baseline scenario, with a modest 2.1% increase in the AFP test and a negligible 0.1% improvement in the AFP 2023 set.
- One drawback of the lemma proposal mechanism is that the model’s performance can be hindered by meaningless proposed lemmas. This issue is evident in the case study section. The key challenge lies in the ability to refine the proposed method to be able to generate lemmas that are genuinely beneficial.

**Questions:**

- Why does the performance of `RL w/o lemma proposal` fare worse than `SFT w/o lemma proposal`? Does the RL loop function as an alternative to expert iteration, serving as an advanced version of it? If so, expert iteration has been shown to be effective for neural theorem proving in previous works such as GPT-f and PACT. Why does the current proposed RL loop fail to improve upon these results?
- Why has the miniF2F result been removed in this submission, despite being presented in the previous NeurIPS submission?

---

> ### Author Response · Authors · 2024-11-19
> **Thank you for the comments.**
>
> We thank Reviewer CLmw for their comments, and for noting “The motivation for this work is strong, addressing a critical problem in the domain of neural theorem proving”. In the following, we address the reviewer’s question in detail.
>
> > Why does the performance of RL w/o lemma proposal fare worse than SFT w/o lemma proposal? Does the RL loop function as an alternative to expert iteration, serving as an advanced version of it? If so, expert iteration has been shown to be effective for neural theorem proving in previous works such as GPT-f and PACT. Why does the current proposed RL loop fail to improve upon these results?
>
> Yes, the RL method is an alternative to expert iteration. Regarding the performance of RL vs. SFT, we note that an important difference between our experiment setups and prior works, such as GPT-f and PACT, is that we let the model generate a complete proof directly without looking at the verifier’s proof state. In the MDP language, the state in our setup is the theorem’s statement and the action is the whole proof, while in GPT-f and PACT, the state is the verifier’s internal proof state (plus some contexts) and the action is a single proof step. Therefore, running RL in GPT-f’s setup can generate data with new MDP states because different proofs may lead to different verifier’s proof states, whereas RL in our setup without lemma proposal cannot generate data with new MDP states. Therefore, the naive RL algorithm on top of the SFT model in our setup cannot collect fundamentally new data, since the ground-truth action for every MDP state in the dataset is already given, and may simply cause overfitting problems. In contrast, with lemma proposals, we can generate data with new MDP states (i.e., the novel lemma statements) and therefore boost the performance of the SFT model.
>
> > Why has the miniF2F result been removed in this submission, despite being presented in the previous NeurIPS submission?
>
> After the NeurIPS deadline, we conducted more extensive parameter tuning and found a better learning rate schedule that significantly improves the performance of both our methods and the baselines (as we pointed out during the NeurIPS rebuttal phase) — we found that models trained with constant learning rates have much better pass@k performance than those trained with cosine learning rate decay because learning rate decay causes more deterministic behavior. As a result, with the new hyperparameters, we do not observe significant improvement on miniF2F compared with the improved performance of the baseline models.

---

> ### Comment · Reviewer_CLmw · 2024-11-23
>
> Thank you for your detailed clarifications.
>
> > Therefore, the naive RL algorithm on top of the SFT model in our setup cannot collect fundamentally new data, since the ground-truth action for every MDP state in the dataset is already given, and may simply cause overfitting problems.
>
> Apologies for not being an RL expert here. Doesn’t the RL process involve sampling additional actions (proofs) for each MDP state (i.e., each problem in the training set) using the current model, and subsequently utilizing these newly sampled actions for RL updates? While I understand that the MDP state (the problems) remains fixed in this context, wouldn’t increasing the diversity of actions used for training improve model performance? From an expert iteration perspective, providing the model with more varied solutions for each problem could potentially enhance performance, even if the effect is less pronounced than in setups like GPT-f.

---

> > ### Author Response · Authors · 2024-11-23
> >
> > We thank the reviewer for the additional comments. Indeed the RL w/o lemma proposal can generate additional actions and increase the diversity of the dataset. However, we believe that the proofs in the AFP files have much higher quality in general since they are written by human experts. In addition, the generated proofs also have a different distribution. Therefore, mixing the generated data may confuse the model because of its limited capacity, or even lower the quality of the dataset.
> >
> > As a more direct comparison, we curated a larger dataset by combining the SFT dataset with all the new proofs generated by the RL w/o lemma proposal algorithm. We trained a new model with the exact same setup as the SFT w/o lemma proposal model. We found that this model performs worse than the original SFT w/o lemma proposal model, with only 42.2% on AFP test and 35.7% on AFP 2023. The results indicate that the generated proofs in general do not enhance the performance of the model.
> >
> > We hope that our response addressed the reviewer’s concerns, and we are happy to answer any follow-up questions.

---

> ### Comment · Reviewer_CLmw · 2024-11-23
>
> Thank you for your further clarification. This is a fascinating and counterintuitive result. Normally, we expect that more diverse proof data would help with the proving process.
>
> I’m now more convinced that this is a really challenging task, and this work demonstrates a solid start in this direction. Current theorem-proving frameworks like AlphaProof have already shown their capability in solving very complex IMO-level problems. Therefore, I believe the direction of attempting the lemma generation task is very important as a future step. I’ve raised my score to 8.

---

### Official Review · Reviewer_kLVh · 2024-11-02

**Soundness:** 3
**Presentation:** 2
**Contribution:** 4
**Rating:** 8
**Confidence:** 4

**Summary:**

According to my understanding, this paper has the following two main contributions:

1. **A Setting for Theorem Proving without the Help of Human-Written Lemmas**

- **Contribution**: This paper introduces a new, more challenging evaluation setting in theorem proving, focusing on the use of human-written code libraries but with minimal reliance on human-proven lemmas.

- **Motivation**: This setting arises from the observation that proofs in human-written projects display strong modularity: proofs are often broken down into lemmas, making final theorems relatively easy to prove by combining these intermediate steps. While efficient, this approach may obscure a prover's ability to handle full-scale proofs of original theorems independently, limiting evaluation of the prover's core capabilities.

- **Method**: To address this, lemmas referenced in the proof of each selected theorem are excluded from the evaluation (however, the authors describe this as removing those "not referred to in the remaining file contents," which seems intended to remove only redundancies). Additionally, the test set is split by dependency—ensuring that no theorem in the test set is refered by any theorem in the training set.

- **Soundness**: Experiments show that the pass rate under the "without lemma proposal" setting is lower than in the "with lemmas  proposal" setting, demonstrating the increased difficulty of this approach.

2. **An Interface for Actively Proposing Lemmas during Proof Generation**

- **Contribution**: This work offers a solution for language models to comply with the above lemma minimalism setting (in my words), enabling models to actively propose lemmas during theorem proving while relying minimally on human-written lemmas.

- **Method**: The theorem proving process is structured as a hierarchical tree search. During proof generation for a specific statement, the model actively proposes lemmas and leverages them in subsequent proof steps, with the proofs of these lemmas deferred to a lower hierarchical level. For model training, reinforcement learning is applied to the supervised-fine-tuned model. By distinguishing between global and local rewards, the model is optimized both to propose provable lemmas that help solve the original theorem and to complete the proofs of proposed lemmas. Special tokens are used to guide the model in deciding when to propose lemmas, effectively controlling exploration.

- **Soundness**: Experimental results indicate that the RL-trained model achieves a higher pass rate than the SFT model, with a 4.7% absolute improvement. Additionally, some proposed lemmas are successfully proved. Experiments over multiple rounds of RL further demonstrate the effectiveness of this approach.

**Strengths:**

1. While the concept of splitting test sets based on file dependencies has been explored in the LeanDojo project ([arXiv:2306.15626](https://arxiv.org/abs/2306.15626)), the minimalist lemma approach introduced here is novel, with comparative experiments validating its challenges.

2. The RL training framework designed to enhance lemma-proposing capabilities is both innovative and practical. The global and local reward design can be generalized to other tree search tasks. Notably, the use of special tokens to control lemma exploration is an exceptional feature.

**Weaknesses:**

1. Although the test set split is restricted by file dependencies, the risk of lemma proposal leakage has not been fully addressed. It is possible that some theorems in the training and test sets rely on the same lemmas, which may have been learned during the supervised-finetuning phase. Furthermore, splitting the test set based on file dependencies might introduce bias. These isolated AFP files could be separated precisely because they are experimental or less widely used, which may not accurately represent the average difficulty of the AFP. The challenges observed in dependency-splitting experiments might instead arise from unfamiliar or unseen knowledge and proving skills specific to these isolated files. An alternative approach would be to train and test on an out-of-domain benchmark, such as miniF2F, which has fewer dependencies and reduces the risk of lemma proposal leakage.

2. The comparison between the SFT and RL models, though straightforward, might be insufficient. It seems that the RL model’s advantages arise from additional exploration and feedback on successful lemma proposals during the online training. A more fair comparison might involve comparing the RL model with an SFT model trained with additional expert iterations, where the SFT model also attempts lemma proposals while without the hierarchical reward.

3. The comparison between proving with and without lemma proposals might also be unfair. In lemma proposal mode, the difficulty of proving theorems is deferred to lemma proving, which has more computational resources than direct theorem proving.

4. *Figure 4* demonstrates that extra lemma proposals may not benefit proofs of higher difficulty. Instead of highlighting the hierarchical approach’s advantages for difficult problems, this approach may exacerbate concerns that its benefits stem from unequal computational budgets.

**Questions:**

1. Regarding *Weakness 4*, the advantages of lemma proposal for more challenging proofs are not immediately clear in *Figure 4*, as the data only presents absolute counts, and the relative advantage is not obvious. Could it be that the relative advantage for proofs with at least 5 steps is indeed more significant?

2. It appears that the tree search depth is limited to 2 or 3. Why have deeper experiments not been conducted? Would increasing the depth help tackle more difficult proofs by allowing finer-grained decomposition, thereby making each lemma more manageable for the model?

---

> ### Author Response · Authors · 2024-11-19
> **Thank you for the comments.**
>
> We thank Reviewer kLVh for their comments, and for noting “the minimalist lemma approach introduced here is novel”, and “RL training framework … is both innovative and practical”. In the following, we address the reviewer’s question in detail.
>
> > It is possible that some theorems in the training and test sets rely on the same lemmas, which may have been learned during the supervised-finetuning phase … An alternative approach would be to train and test on an out-of-domain benchmark, such as miniF2F, which has fewer dependencies and reduces the risk of lemma proposal leakage.
>
> We test on the miniF2F benchmark and find that our method does not have significant improvement over the baselines. We hypothesize the reasons to be: (a) the theorems are very different from theorems in the training dataset, and (b) we also observe that when tested on miniF2F theorems, our model failed to propose meaningful lemmas. This may be because proving to miniF2F-level mathematics questions typically does not require hierarchical decomposition. In contrast, the AFP datasets work with abstract mathematical objects and theorems, which intrinsically have hierarchical decompositions.
>
> We also agree with the reviewer that, even if we split the training and test set by the file dependencies, or even by the submission date like our AFP 2023 test set, theorems in the training and test set may rely on the same lemma. Such dependency cannot be eliminated easily even if we test on the miniF2F benchmark because the proof still depends on some basic lemmas such as the properties of prime numbers.
>
> Nonetheless, we believe that our train/test split serves as a step toward building rigorous benchmarks for testing the fundamental reasoning capabilities of LLMs.
>
> > Regarding Weakness 4, the advantages of lemma proposal for more challenging proofs are not immediately clear in Figure 4, as the data only presents absolute counts, and the relative advantage is not obvious. Could it be that the relative advantage for proofs with at least 5 steps is indeed more significant?
>
> We are not sure whether we understand the reviewer’s question correctly (please kindly let us know if not). In Figure 4, we show the pass rate of the theorems grouped by the depth of their ground-truth proof trees. Taking the right-most stacked bar as an example, there are 336 theorems with ground-truth depth >= 4 in the test set, and our model proves 14 of them. Therefore the pass rate is 4.17% as shown in Figure 4. Among the 14 proved theorems, 1 of the theorems is proved with a proof tree of depth 2, and 1 of the theorems is proved with a proof tree of depth 3. Similarly, the baseline model proves 10 theorems with direct proofs only, which translates to a pass rate 2.98%. The relative advantage for theorems with ground-truth depth >= 4 is then 40%, subject to a non-negligible statistical error due to the small sample size.
>
> > It appears that the tree search depth is limited to 2 or 3. Why have deeper experiments not been conducted? Would increasing the depth help tackle more difficult proofs by allowing finer-grained decomposition, thereby making each lemma more manageable for the model?
>
> We thank the reviewer for the suggestion. First, at the risk of being redundant, we would like to point out that our current algorithm does not involve tree search. We construct the proof tree merely by sampling the proof for each proposed lemma once. While combining our proof tree structure with search algorithms such as MCTS is an interesting question, we leave it as a future work and mostly focus on the effect of proposing new lemmas during training in this paper.
>
> While allowing finer-grained decomposition may make each lemma easier, proposing more lemmas may not always increase the pass rate of the model due to the definition of global correctness — a theorem is proved only if all the proposed lemmas are also proved. As a concrete example, even if each proposed lemma can be proved with probability 50%, with tree depth 5, the overall pass rate drops to about 3% even if only one lemma is proposed in every conditional proof. This is consistent with our empirical observation in Figure 4, where only a very small amount of theorems can be proved with tree depth > 3. To handle deeper proof trees, we might need a fundamentally better model by scaling up either the model size or the dataset, which are unfortunately out of our compute budget.

---

> > ### Comment · Reviewer_kLVh · 2024-11-19
> > **Further Remarks and Score Revision**
> >
> > Thank you for your detailed discussion. While there are many aspects that could be further developed, the paper deserves acceptance for its outstanding originality. Below are some additional comments.
> >
> > > (b) we also observe that when tested on miniF2F theorems, our model failed to propose meaningful lemmas. This may be because proving to miniF2F-level mathematics questions typically does not require hierarchical decomposition.
> >
> > I generally disagree with this hypothesis. In fact, miniF2F includes challenging competition problems sourced from AMC, AMIE, and IMO, which often exhibit clear hierarchical decompositions of immediate conclusions in their natural language solutions. Therefore, I suspect this failure is due to the model’s inability to fully learn how to actively decompose complex mathematical problems and plan sequential solution steps, rather than simply imitating patterns demonstrated in AFP.
> >
> > However, this limitation generally does not stem from any shortcomings on the part of the authors but instead reflects the broader need for progress in training LLMs for both formal and natural language mathematics. Considering the paper’s significant contributions in terms of motivation and methodology, I have raised my score to 8 (accept).

---

> > > ### Author Response · Authors · 2024-11-21
> > > **Thank you for the comments!**
> > >
> > > We thank reviewer kLVh for the additional comments, and for raising the score.
> > >
> > > > I suspect this failure is due to the model’s inability to fully learn how to actively decompose complex mathematical problems and plan sequential solution steps, rather than simply imitating patterns demonstrated in AFP.
> > >
> > > This is a good point! Decomposing theorems from the miniF2F dataset could be quite different from decomposing theorems in AFP, as the difficulty of math competition problems is also somewhat different from the difficulty of the abstract theorems in AFP. It is indeed possible that the decomposition ability of our model does not generalize well to miniF2F. We will include a detailed discussion regarding this point upon revision of the paper.

---

### Meta-Review · Area_Chair_BczD · 2024-12-26

**Metareview:**

This paper focuses on neural theorem proving in contexts where the relevant premises are unavailable. The authors propose Proof Decomposer (ProD), a RL approach that utilizes hindsight experience replay to reward a LLM in hierarchically decomposing a given theorem into lemmas, and subsequently proving these lemmas to prove the theorem. Experiments on the AFP dataset indicate that the proposed method (ProD-RL) outperforms several baseline approaches, including those relying solely on supervised fine-tuning (SFT), as well as methods without lemma proposal.

Reviewers generally agree that the paper is well-written (Reviewers CLmw, vahD) and find the new task both challenging and interesting (Reviewers kLVh, CLmw). They also note that the method is well-motivated (Reviewers kLVh, CLmw). However, as pointed out by Reviewers CALz and vahD, ProD shares notable similarities with existing methods, albeit in a slightly different setting. A key point raised by multiple reviewers (CLmw, CALz, vahD) is the lack of experiments on the miniF2F dataset, which appeared to be selectively omitted.

In summary, while the motivation and setup of the proposed method are compelling, the novelty and especially the generalization performance of ProD-RL remain concerns. The authors are encouraged to conduct further experiments to strengthen generalization on miniF2F and to include these extended results in the revised paper.

**Additional Comments On Reviewer Discussion:**

During the rebuttal, the authors address most of the concerns raised by reviewers kLVh and CLmw and provide additional experiments on the miniF2F dataset (Reviewers CALz, vahD). However, ProD-RL underperforms compared to basic SFT on miniF2F, revealing limited generalization capabilities—an issue of critical importance. The authors also present additional experiments leveraging ProD-RL in another setting, yet the ProD-SFT-aug approach either underperforms or only matches the performance of basic SFT methods. These results underscore the need for further improvements to enhance the method’s generalization performance.

---

### Decision · Program_Chairs · 2025-01-22

Reject